# Efficacy of New Local Bacterial Agents against *Pyrenophora tritici-repentis* in Kuban Region, Russia

**Anzhela Asaturova \*, Natalya Zhevnova, Natalya Tomashevich** , **Marina Pavlova, Oksana Kremneva** , **Galina Volkova and Nikita Sidorov**

Federal Research Center of Biological Plant Protection, 350039 Krasnodar, Russia; nataliaznevnova@gmail.com (N.Z.); nataliatomashevich@yandex.ru (N.T.); fridaamely@yandex.ru (M.P.); kremenoks@mail.ru (O.K.); galvol.bpp@yandex.ru (G.V.); elisitor@mail.ru (N.S.)

\* Correspondence: tiamat-7@mail.ru; Tel.: +7-9180801572

**Abstract:** The phytopathogenic fungus *Pyrenophora tritici-repentis* is a causal agent of tan spot. Antagonistic microorganisms can be used as a non-chemical alternative treatment against the tan spot of wheat. *Bacillus velezensis* BZR 336 g and BZR 517 stains were selected as the most active microorganisms and potential biocontrol agents. We found that *B. velezensis* strains BZR 336 g and BZR 517 exhibited antagonistic activity against *P. tritici-repentis* Kr-15/2016 in vitro: they inhibited mycelium growth by 72.4–94.3% and caused its degenerative changes. Treatment of seeds and plants with strains BZR 336 g and BZR 517 provided a biological efficiency of 31.2–38.4% against tan spot, while artificial inoculation of plants provided only 28.4–43.8% biological efficiency. Treatment of seeds and plants with BZR 336 g and BZR 517 in a three-year field trial demonstrated 24.6–50% biological efficiency. BZR 336 g and BZR 517 provided 5.0–7.6% additional yield. We conclude that BZR 336 g and BZR 517 are promising options for novel bioproducts that can control *P. tritici-repentis* tan spot.

**Keywords:** *Bacillus velezensis*; *Pyrenophora tritici-repentis*; biological plant protection; tan spot; winter wheat; microbial strains

## 1. Introduction

Tan spot, also known as yellow leaf spot, is one of the major diseases of winter wheat (*Triticum aestivum* L.) and is caused by the fungal pathogen *Pyrenophora tritici-repentis* (Died.) Drechs (anamorph of *Drechslera tritici-repentis* (Died.). This pythopathogen attacks wheat leaves causing necrotic lesions and chlorotic spots. Consequently, the plant photosynthetic area decreases, resulting in leaf death and, finally, in reduced leaf quality [1,2]. Yield losses can reach up to 50% [3].

Tan spot occurs worldwide in all major wheat growing regions [4]. The disease has been documented in Syria, Argentina, Brazil, Algeria, the USA, Canada, Russia and some other countries [2,5–8].

Key factors of its occurrence include minimal tillage and short crop rotation [8–10], continuous wheat cropping and the use of susceptible varieties [1,11], as well as excessive use of chemical pesticides [10,11]. Populations of *P. tritici-repentis* have a high genetic diversity, which positively affects the host range and virulence [12]. *P. tritici-repentis* conidia can survive for a long time on plant debris and spread over large distances [5].

In Russia, tan spot was identified in 1985 in the North Caucasus and subsequently spread throughout the country. Today, this problem remains relevant for the Kuban Region (Northern Caucasus, Russia), where wheat is the main cultivated crop, and farmers seek to reduce the number of chemical treatments. In recent years, this disease has seriously affected wheat production in terms of quantity and quality [10,13].

The strategy of combating tan spot disease is complex and consists of a combination of chemical, cultural, genetic, and sometimes biological control measures [1]. Currently, chemical pesticides are mainly used. However, they negatively affect the environment and

human and animal health [8]. The overuse of fungicides can make the fungicide pathogen-resistant [7,14]. Moreover, sometimes chemicals are not effective against *P. tritici-repentis* [15]. Eco friendly plant growth-promoting rhizobacteria (PGPR) and biopreparations based on them are an alternative to chemical fungicides. Numerous reports show that PGPRs are effective biocontrol agents that reduce plant diseases and can directly or indirectly stimulate plant growth and development [6,10,11,16].

Bacteria from the *Bacillus* group are effective bioagents against fungal plant pathogens. Bacteria influence many growth- and immunity- related metabolic processes in the plant. *Bacillus* can produce lots of secondary metabolites of various chemical natures. This determines their suppressive properties in relation to a wide list of phytopathogens. Cyclic lipopeptides play a key role in the induction of plant immunity due to microorganisms. These substances provide protection against a wide range of pathogens through antagonistic and/or induced resistance effects [17].

*B. velezensis* produces a large number of antimicrobial metabolites: lipopeptides, polypeptides, enzymes, and non-peptide compounds [3–7]. The mechanism of action (MOA) of lipopeptides on filamentous fungi is associated with their effect on membranes through interaction with ergosterol. As a result, pores are formed, followed by the release of monovalent cations from the cells, which leads to lysis.

*B. velezensis* strains associated with winter wheat plants may be of interest as biological agents for tan spot control and increasing crop yields. The bacteria have been proven to be effective on various crops and in different climate regions. *B. velezensis* showed great antagonist activity in the south of Russia [18], as well as in the extreme weather of Western Siberia [19,20]. These microorganisms promote plant growth [21], making them excellent candidates for the development of the new bioproducts.

Data analysis of the literature shows insufficient information on the effectiveness of bacteria of the genus *Bacillus* against tan spot. Thus, in sustainable agriculture, there is a timely need for promising strains of microorganisms, along with the development of microbial based biocontrol agents against tan spot. The use of PGPRs and other microbial-based products is relevant for agriculture, especially for organic farming, where the use of chemical pesticides is prohibited.

In this research, we aim to study the interaction mechanism of *Bacillus* biocontrol strains with a tan spot pathogen (*P. tritici-repentis*), as well as to study the biological effectiveness of environmentally friendly microbiological agents under conditions of natural infection.

## 2. Materials and Methods

### 2.1. Bacterial and Fungal Strains

In the research we used two initial strains of *Bacillus* isolated from the rhizosphere of winter wheat in Krasnodar (Kuban Region, Krylovsk district 46°19′19″ N 39°57′51″ E; Pavlovsky district 46°08′22″ N 39°47′19″ E).

Through genome sequencing of these strains, we identified their taxonomic position. Comparative bioinformation analysis resulted in the collection and deposition of two genomes into NCBI database: *B. velezensis* BZR 336 g_n (Assembly: GCA_009683125.1, GenBank: NZ_WKKU00000000.1) and *B. velezensis* BZR 517_n (Assembly: GCA_009683155).1, GenBank: NZ_WKKV00000000.1) [21,22].

The plants were inoculated with the *P. tritici-repentis* isolate Kr-15/2016 recovered from the susceptible winter wheat variety Batko. The *P. tritici-repentis* Kr-15/2016 isolate was typed to race 8 based on its ability to induce necrosis and chlorosis in cv. Glenlea and the 6B365 and 6B662 lines [23]. This isolate produces three toxins: Ptr ToxA, Ptr ToxC, or Ptr ToxB [24,25]. We chose this race because it produces all three known toxins and is widespread in Krasnodar Krai [26].

Strains of bacteria and fungi belong to the unique scientific facility State Collection of Entomoacariphages and Microorganisms of the Federal Scientific Center for Biological Plant Protection, Krasnodar, Russia (http://ckp-rf.ru/ no. 585858, accessed on 2 December

2021). Biocontrol strains have passed toxicological and hygienic examination at the Center for Toxicology and Hygienic Regulation of Biopreparations (Serpukhov, Russia). In the research we used the Unique Scientific Database «Technological line for obtaining microbiological plant protection products of a new generation» (https://ckp-rf.ru/usu/671367/ accessed on 2 December 2021) of the Federal Scientific Center for Biological Plant Protection, Krasnodar, Russia.

For each experiment, fresh liquid cultures were prepared on the original optimized nutrient medium in New Brunswick Scientific Excella E25 (Enfield, USA) shaker-incubators (180 rpm). The initially optimized culture medium was obtained with a modified Czapek medium containing corn extract (nitrogen source) and molasses (carbon source). The incubation period of the BZR 336 g strain was 48 h at 25 °C; BZR 517 strain—36 h at 30 °C [26].

### 2.2. Laboratory Experiments

Antagonistic activity was examined with the dual cultures method on optimized culture media [27]. We placed mycelial agar blocks in Petri dishes (PD), while the bacterial strain was applied by the stroke method at a distance of 6 cm from the pathogen block. The cultures were then incubated for 20 days at 28.0 °C. Pure pathogen fungal cultures and bacteria seeded separately are used as controls.

The counts were carried out on the fifth and fifteenth days. The degree of fungal mycelium inhibition was calculated by the following formula [28]:

$$I, \% = (Dc - Dt)/Dc \times 100$$

where Dc is the diameter of fungal colony in the control, and Dt is the diameter of fungal colony in the dual culture.

We examined bacterial-fungal interactions with a Zeiss AxioScope A1 microscope (Carl Zeiss, Oberkochen, Germany). When filling an optimized culture media in PDs, the bacterial culture was placed on a glass slide on one side, and *P. tritici-repentis* mycelium on the other. PDs then were incubated at 24–26 °C under light. The slides were not stained [29].

We observed the pathogen-antagonist system, as well as length and diameter measurements of mycelium segments with a Carl Zeiss AxioVision Rel. 4.8.2 (Carl Zeiss, Moscow, Rusia) software in dynamics on the third, seventh, tenth and fourteenth days of co-cultivation.

### 2.3. Growth Chamber Assay

We sought to determine the effect of pre-sowing treatment with bio-agents on the resistance of winter wheat plants to *P. tritici-repentis* Kr-15/2016, despite the fact that tan is an aerogenic infection. Consequently, we provided for two variants of the experiment: (1) presowing treatment followed by the treatment of vegetative plants; (2) treatment of vegetative plants. In the experiment, seeds of soft winter wheat variety Batko were used. This variety is highly susceptible to tan spot—the incidence rate is up to 60%.

The experiment consisted of four stages: presowing treatment (or sowing grain without treatment), first treatment of vegetative plants, pathogen suspension inoculation, second treatment of vegetative plants (Figure 1).

It is common to use V-8 juice to grow fungi. It is a blend of 8 vegetable juices. This mixture is not produced in Russia. Therefore, we used a V-4 nutrient medium, consisting of a 15% mixture of vegetable juices in a ratio of 4:3:2:1 of beetroot, parsley or celery, carrot and tomato juices, respectively, at 0.3% $CaCO_3$, 2% agar. This nutrient medium contributes to the abundant sporulation of the fungus.

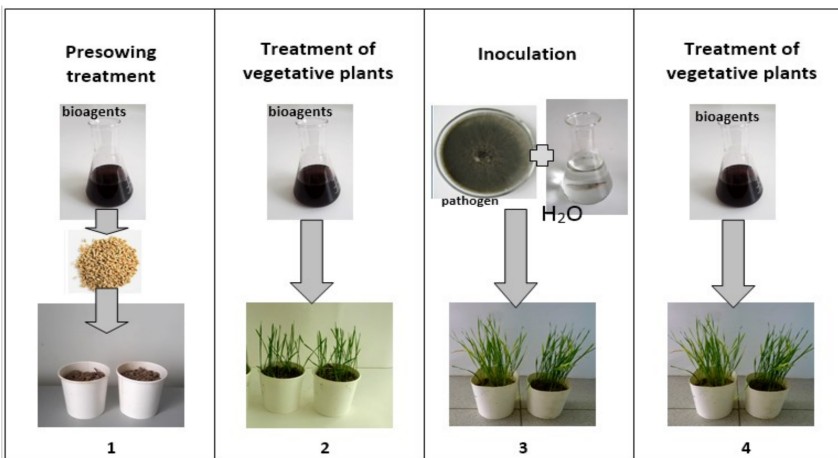

**Figure 1.** Stages experiment in climate chamber.

The fungus was grown on V-4 solid nutrient medium for three days at 25 °C. The culture was then exposed to UV for five days to stimulate the formation of conidiophores. Then the fungus was placed in a refrigerator for one day to form conidia. A pure culture of the spore fungus *P. tritici-repentis* Kr-15/2016 was used on a solid nutrient medium nine days old to prepare a water-conidial suspension. Mycelium of fungi was scraped off the surface of the nutrient medium. We homogenized mycelium papillae in 100 mL of sterile distilled water to obtain fungal spore suspensions. Conidia number per unit was counted in the Goryaev chamber. The final spore suspension concentration was $3–5 \times 10^3$ spores/mL; the infectious load was 50 mL/m$^2$ [26]. Our experimental research determined that this concentration promotes the optimal level of infection.

Seeds were sown in glasses with sand. There were 30 seeds in one glass. All experiments were performed in triplicate. Plants were germinated in a climatic chamber (Binder KBWF, Tuttlingen, Germany) at a temperature of 25 °C, illumination of 14,000 lux, and humidity of 40% for six days until the double leaf phase and then were infected with a pathogen.

Before inoculation, we carefully removed the epicuticular wax coating from the surface of the leaves and treated them with a water-conidial suspension of the fungus *P. tritici-repentis* Kr-15/2016 using a spray gun. The removal of epicuticular wax from the leaf surface facilitates better penetration of the pathogen into the plant. By artificially stimulating the adaptive immune defenses we aim to get more pronounced disease symptoms. The infected plants were incubated for 16 h at 100% humidity and 20 °C in the dark.

After inoculation, the plants were grown in a climate chamber at 25 °C, 14,000 lux illumination, and 40% humidity for five days. Watering was done as needed. Vegetative plants were treated with laboratory samples of the liquid cultures twice: the first was on the third day after sowing; and (2) three days after inoculation. The fungus had already penetrated the plant and/or started to develop during the second treatment. Still, we wanted to establish whether the disease was curable at this stage.

The pesticide Raxil, KS B (Bayer, Moscow, Russia) was used as a chemical standard for seed treatment (tebuconazole, 60 g/L, application rate 0.45 L/t); Alto Super, KS (propiconasol, 250 g/L, cyproconazole, 80 g) (Syngenta, Moscow, Russia) was used for the treatment of vegetative plants (application rate 0.45 L/ha). In the control, seeds and plants were treated with distilled water.

In this research, we determined biological efficacy by two criteria: the number of infection points and the type of sustainability. This is the type of sustainability determined by the method of Rees et al. [30]. This scale takes into account the size and type of lesions (Table 1).

**Table 1.** Scale for assessing the resistance of wheat varieties to the yellow leaf spot pathogen in the seedling phase (Rees, Platz, 1987).

| Size and Type of Lesion | Reaction Type, Point | Wheat Phenotype |
|---|---|---|
| No symptoms | 0 | HR |
| Small (up to 0.5 mm) dark brown spots. There are no chlorosis or they are small | 1 | R |
| Dark brown spots up to 1 mm. There may be chlorosis. | 2 | MR |
| Small spots (1–2 mm) from pale to dark brown, often with a yellow halo | 3 | MS |
| Large (3 mm) brown spots, usually with a dark brown center. Mostly surrounded by significant chlorosis from 2 to 3 mm. | 4 | S |
| Large (3–5 mm) necrosis with a dark brown center, severe yellowing of the surrounding tissues. The spots coalesce, resulting in the death of part or all of the leaf. | 5 | HS |

Note. HR—high resistance, R—resistance, MR—medium resistance, MS—medium susceptibility, S—susceptibility, HS—high susceptibility.

The number of infection points shows the ability of a bioagent to inhibit the penetration of a pathogen into a plant, and the type of sustainability shows the ability of a bioagent to restrain the development of the infection that has already penetrated into plant tissue.

We applied Abbott's formula to distinguish biological efficiency by the number of foci of infection and the type of resistance [31]:

$$BE = (100 \times (P - p))/P,$$

BE—biological efficiency, %; P—infection intensity in the control, %; p—infection intensity with treatment, %.

*2.4. Field Trials*

Field trials were held in 2013–2016 on the experimental plots of the Federal Scientific Center for Biological Plant Protection (Russia, Krasnodar, 45°02′57.2″ N 38°52′29.7″ E) under natural field conditions. Sunflower was a test plant in the 2013–2014, 2015–2016 growing seasons; burclover was tested in 2014–2015. An eight-field crop rotation system was used.

Sowing was postponed for three weeks due to adverse weather in 2015–2016. The total area of each option was 500 m$^2$. The winter wheat variety Kalym was used (mid-season, semi-dwarf, highly resistant to lodging, leaf rust, septoria, and powdery mildew). This variety had a medium resistance to *P. tritici-repentis* Kr-15/2016 (the incidence rate was up to 30%) and is widespread in the Kuban region.

The neighboring experimental plots with a large amount of plant residues were the primary source of infection. In addition, the weather contributed to its spread. As the plants grew, the infection intensity increased due to both expansion of the previously infected spots and the appearance of the new ones. This indicates re-infection during the season.

Seeds were treated before sowing; vegetative plants were treated twice. The liquid crop application rate of BZR 336 g and BZR 517 was 3 and 2 L/t respectively for seed treatment; and 3 and 2 L/ha for spraying plants. The working fluid consumption was 10 L/t and 300 L/ha. In the control, the seeds were treated with tap water. Sowing was carried out with a mechanical seeder SZN-3.6 (Russia). The seeding rate was 220 kg/ha, with row spacing of 15 cm. The grain was treated with a manual sprayer and mixed manually using agricultural equipment; the plants were treated with a mounted boom sprayer.

We applied Stomp herbicide, CE (pendimetalin, 330 g/L) at a rate of 4.5 L/ha prior to the emergence of plant shoots. In the spring, the herbicide Prima SE (300 g/L 2,4-D (2-ethylhexyl ether) + 6.25 g/L florasulam) was used at an application rate of 0.6 L/ha in the tillering phase.

The severity of leaf disease was evaluated sequentially as it spreads. We visually assessed the leaf damage area as a percentage to the total leaf area in the tillering phase in autumn (Z 20–21) and spring (Z 26–21) at the shooting stage (Z 32–35) and at maturation (Z 73–77). The data in the table were obtained at the maximum development of the disease. We applied Abbott's formula to calculate biological efficiency [31].

### 2.5. Statistical Analysis

All experiments were performed in triplicate. We ran Duncan's Multiple Range Test in STATISTICA 13.2 EN (trial version, Tibco, Palo Alto, CA, USA).

## 3. Results
### 3.1. Study Design

The research work consisted of laboratory studies (analysis and microscopy of dual cultures), studies in a climatic chamber of artificial infection under controlled conditions, and three-year field trials of natural infection.

### 3.2. Laboratory Experiments

The in vitro strains BZR 336 g and BZR 517 exhibited good antagonistic properties against *P. tritici-repentis* Kr-15/2016. Maximum growth suppression of the pathogen mycelium reached 93.9%. BZR 336 g in high mobility provided a noticeable antifungal effect (81.8–94.3%). This resulted in the rapid decrease of the sterile area. The *B. velezensis* BZR 517 strain did not show such mobility. Its antifungal activity ranged from 57.6 to 72.4%. However, this strain retained the sterile area for a long time (Table 2, Figure 2).

**Table 2.** Antifungal activity of BZR 336 g and BZR 517 against *P. tritici-repentis* Kr-15/2016 (the dual culture method).

| Co-Cultivation Options | The Growth of Mycelium from the Seed Block, mm | | Inhibition of Mycelium Growth, % | |
|---|---|---|---|---|
| | 5th Day | 15th Day | 5th Day | 15th Day |
| Control (*P. tritici-repentis* Kr-15/2016) | 13.2 [c] | 60.9 [c] | - | - |
| BZR 336 g | 2.4 [b] | 3.5 [a] | 81.8 | 94.3 |
| BZR 517 | 5.6 [a] | 16.8 [b] | 57.6 | 72.4 |

Note: Different letters indicate significant differences ($p < 0.05$) according to Duncan's test.

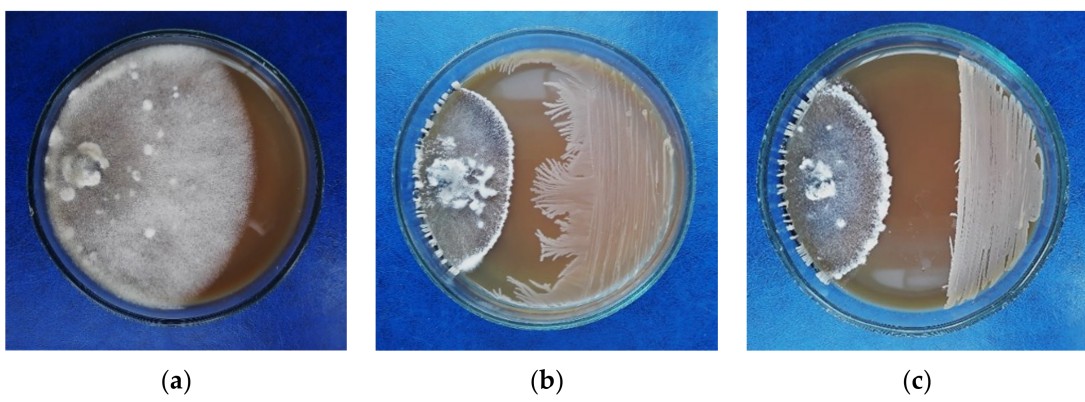

(**a**)          (**b**)          (**c**)

**Figure 2.** Dual cultures of *P. tritici-repentis* Kr-15/2016 and antagonistic strains: (**a**) control without an antagonist, (**b**) together with BZR 336 g, (**c**) together with BZR 517; seven days of co-cultivation.

*P. tritici-repentis* mycelium segments were shortened by 2.5–4 times and thickened by 2–2.5 times compared with the control on the third day of joint cultivation with the strains BZR 336 g and BZR 517. The growth mycelium in terms of length was noticeably inhibited, and its branching increased (Figure 3).

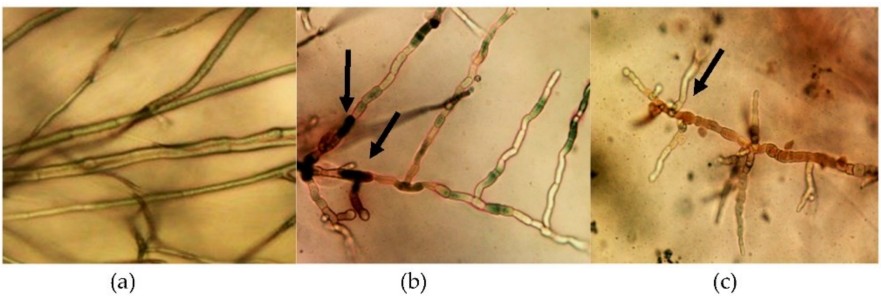

(a)           (b)           (c)

**Figure 3.** Hyphae of *P. tritici-repentis* Kr-15/2016: in control without an antagonist (**a**); after three days of co-cultivation with *B. velezensis* BZR 336 g (**b**) and BZR 517 (**c**), arrows show the pigmentation of hyphae; ×200 magnification.

We observed "vacuolization" (the formation of large membranous vesicles) in the cytoplasm of pathogen hyphae on the 3rd–7th day of BZR 336 g strain fungus cultivation (Figure 4).

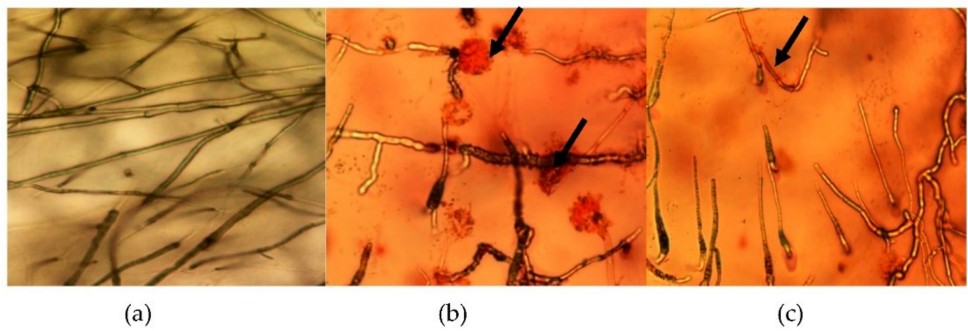

(a)           (b)           (c)

**Figure 4.** *P. tritici-repentis* Kr-15/2016 hyphae: control (**a**); and vacuolization of hyphae after 3 days of co-cultivation with strain BZR 336 g (**b**) and on the seventh day of co-cultivation with strain 517 (**c**); ×400 magnification.

On the 7–10th day, the pathogen mycelium broke up into separate fragments releasing the pigmented hyphae content. We observed a similar process at BZR 517 strain cultivation, though with a delay of three to four days (Figure 5).

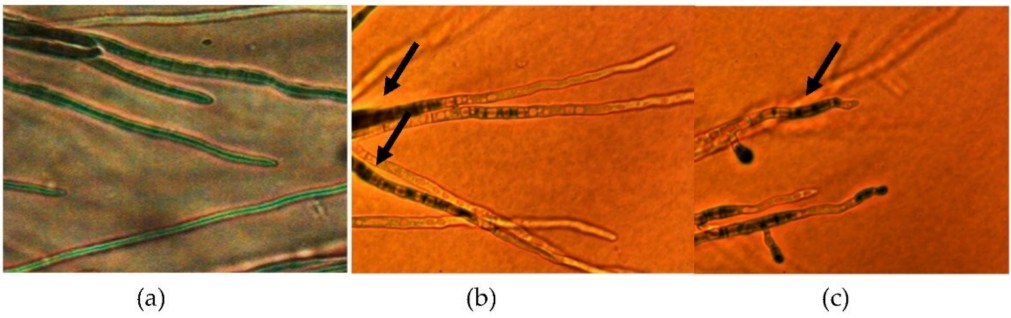

(a)           (b)           (c)

**Figure 5.** Hyphae of *P. tritici-repentis* Kr-15/2016: control (**a**); decomposition of mycelium during co-cultivation with strain BZR 336 g on day 10 (**b**), and with 517 strain on day 14 of co-cultivation (**c**). Arrows show the areas with the release of pigmented hyphae contents; magnification ×200.

We did not register direct contact between the fungus and bacteria. Nevertheless, we noted a chemotaxis of BZR 336 g bacterial cells towards the fungal hyphae, after which the hyphae completely dissolved. BZR 517 cell movement was insignificant or absent.

### 3.3. Growth Chamber Assay

Two criteria determined the biological efficiency: (1) the number of infection points (bioagent ability to suppress the penetration of the pathogen into the plant); (2) type of resistance (bioagent ability to control an infection which has already penetrated the plant tissue). The biological efficiency of the liquid culture based on BZR 336 g with seed treatment was 38.4% in terms of reducing the number of foci of infection and 50.8% in terms of resistance type. It was 10 and 38% higher than when only treating the plants (Table 3).

**Table 3.** Biological efficacy (BE, %) of strains BZR 336 g and BZR 517 against *P. tritici-repentis* Kr-15/2016 under conditions of artificial inoculation of winter wheat Batko in the climatic chamber.

| | | Methods of Treatment | | | |
| --- | --- | --- | --- | --- | --- |
| | | Control | Chemical Treatment | BZR 336 g | BZR 517 |
| Seed and plant treatment | the number of infection points | 2.37 [b] | 0.44 [c] | 1.48 [a] | 1.63 [ab] |
| | type of sustainability, score | 1.85 [be] | 0.63 [ab] | 0.91 [cd] | 1.48 [ab] |
| | BE by the number of infection points, % | - | 81.4 | 38.4 | 31.2 |
| | BE by of sustainability, % | - | 65.9 | 50.8 | 20.0 |
| Plant treatment | the number of infection points | 5.25 [b] | 1.40 [c] | 3.76 [a] | 2.95 [a] |
| | type of sustainability, score | 2.55 [ac] | 1.03 [b] | 2.22 [a] | 1.40 [b] |
| | BE by the number of infection points, % | - | 73.3 | 28.4 | 43.8 |
| | BE by of sustainability, % | - | 59.6 | 12.9 | 45.1 |

Note: Different letters indicate significant differences ($p < 0.05$) according to Duncan test.

A liquid culture based on the BZR 336 g strain turned out to be more effective in the treatment of seeds and plants—50.8% in terms of resistance type and 38.4% in terms of infection points. Efficiency was lower when treating only plants. In turn, the liquid culture based on the BZR 517 strain proved to be highly effective only for plant treatment: 43.8% in terms of infection points and 45.1% in terms of resistance type. When seeds were treated with this strain, its efficiency decreased by 13–25% (Table 3). The effectiveness of the standard in the treatment of seeds and plants was high: 81.4% by the number of infection points and 65.9% in terms of resistance type ($p = 0.05$). However, these figures were only 6–8% higher than when treating plants alone.

### 3.4. Field Trials

The incidence rate of tan spot in the control was 3.3% in the growing season of 2013–2014. A statistically significant reduction in the severity of the disease was observed when using *B. velezensis* BZR 336 g, *B. velezensis* BZR 517 and a chemical standard ($p < 0.05$). The standard proved the highest efficiency of 83.1% (Table 4).

The severity of tan spot was 7.5%. In the growing season of 2014–2015, all experiments registered a statistically significant reduction in the incidence rate ($p = 0.05$). The biological efficiency of the strains was 41.8–43.6%. The effectiveness of the chemical standard was 83.7% (Table 4).

In 2015–2016, the incidence rate was 7.6%. All experiments registered a statistically significant reduction of the tan spot rate ($p = 0.05$). The liquid culture efficiency for BZR 336 g was 50.0% and for BZR 517 it was 34.2%. The biological effectiveness of chemical fungicide was 65.7% (Table 4).

**Table 4.** Biological efficacy (BE, %) and yield of winter wheat of Kalym cultivar treated with the strains BZR 336 g and BZR 517 under stationary crop rotation, 2013–2016.

|  | Methods of Treatment | Severity, % | BE, % | Yield, t/ha |
|---|---|---|---|---|
| 2013–2014 | Control | 3.3 [d] | - | 6.9 [a] |
|  | Chemical treatment | 0.6 [c] | 83.1 | 7.0 [a] |
|  | BZR 336 g | 2.0 [ab] | 49.2 | 7.2 [c] |
|  | BZR 517 | 1.7 [ab] | 38.5 | 7.6 [b] |
| 2014–2015 | Control | 7.5 [b] | - | 6.0 [a] |
|  | Chemical treatment | 1.2 [c] | 83.7 | 7.1 [ab] |
|  | BZR 336 g | 4.4 [a] | 41.8 | 7.4 [b] |
|  | BZR 517 | 4.2 [a] | 43.6 | 7.1 [ab] |
| 2015–2016 | Control | 7.6 [b] | - | 5.2 [ab] |
|  | Chemical treatment | 2.6 [a] | 65.7 | 5.6 [c] |
|  | BZR 336 g | 3.8 [a] | 50.0 | 5.0 [a] |
|  | BZR 517 | 5.0 [a] | 34.21 | 5.5 [bc] |

Note: Different letters indicate significant differences ($p < 0.05$) according to Duncan's test.

Both in 2013–2014 and 2014–2015 the yield in the control was approximately the same (6.9–6.0 t/ha and 6.4–7.4 t/ha, respectively). We recorded a statistically significant increase in yield above the chemical standard using BZR 336 g (7.2 t/ha in 2014 and 7.3 t/ha in 2015) and BZR 517 (7.6 t/ha in 2014 and 7.1 t/ha in 2015) (Table 4).

We had the lowest yield in 2015–2016 as a result of shifting the planting dates by three weeks due to unfavorable weather. In the control it was 5.2 t/ha. Only the standard indicated a statistically significant (5.7 t/ha) increase in yield compared to the control (Table 3).

## 4. Discussion

Biological control is a promising plant pathogen control strategy. The findings of previous studies confirm the effectiveness of antagonistic microorganisms as control phytopathogens [3,6,10,11,16,32].

Several studies consider the interaction of *P. tritici-repentis* with antagonistic PGPR [6,10,11,16,32]. Larran et al. [10] showed that endophytic microorganisms isolated from wheat, including *Bacillus* sp., proved to be effective biocontrol agents for *P. tritici-repentis*. They suppressed the growth of pathogens, reduced the diameter of fungal colonies, the percentage of spore germination in the laboratory, and disease severity in the greenhouse. Microscopic examination revealed degenerative changes in the mycelium of pathogenic microorganisms during co-incubation with isolates of bacteria and fungi. Researchers observed conidial plasmolysis, shortening and swelling of germ tubes, hyphae vacuolization, induced chlamydospore formation, and production of pigmented compounds within hyphae or in the culture medium [10].

In Perello et al. [6], bioagents for pre-sowing seed/vegetative plants treatment reduced the incidence of tan spot from 16 to 56%. Syrian researchers studied the effect of promising strains on susceptibility to tan spot in barley varieties differing in resistance to *P. graminea*. They proved a significant increase in plant resistance [11]. Pfender et al. [32] described the reduction of primary inoculum of a phytopathogen in surface crop residues when microbial bioagents were applied.

This paper is a part of a comprehensive study of the effect of bacterial strains BZR 336 g and BZR 517 against winter wheat diseases. Previously, we used these strains to control *Fusarium* root rot disease, but then we discovered that they are effective against leaf diseases as well [18,33].

Here we found that *B. velezensis* strains BZR 336 g and BZR 517 showed antagonistic activity against *P. tritici-repentis* in vitro and caused degenerative changes to the pathogen mycelium. We assume that BZR 336 g possesses a MOA based on active mobility, while

BZR 517 is based on active production of antifungal metabolites. Future research will need to clarify what classes these substances belong to.

We concluded that BZR 336 g and BZR 517 were effective against *P. tritici-repentis* in a climate chamber. Two criteria determined biological efficacy: (1) bioagent ability to suppress the penetration of the pathogen into the plant; and (2) bioagent ability to control an infection which has already penetrated the plant tissue [30]. The biological efficiency of the treatment of seeds and plants with BZR 336 g was 10% higher than when treating the plants alone. In turn, BZR 517 proved to be highly effective only in plant treatment. Presumably the reason lies in different MOA of strains on the pathogen.

We conclude that the sequential treatment of seeds and plants is most effective. A. Perello [6] and W.C. Luz [16] described the method of seed treatment with microorganisms to control *P. tritici-repentis*. The authors assumed that seed treatment is most effective in reducing the incidence of tan spot. Apparently, this treatment causes systemic resistance responses in plants.

In our study, biological efficiency did not correlate with yield. For example, in 2013–2014 the protective effect of BZR 336 g and BZR 517 on plants was lower than that of the chemical fungicide (standard). However, the additional yield was higher than in the standard. The yield in the control was 6.9 t/ha.

In 2015–2016, we received low yields on the field, as a result of shifted planting dates due to unfavorable weather. A statistically significant increase in yield compared to the control was obtained only when using the chemical standard (5.6 t/ha) (Table 3). Our results prove that living microorganisms are very sensitive to environmental conditions; therefore, strict adherence to agrotechnological measures is necessary.

## 5. Conclusions

The outcome indicates that *B. velezensis* strains are effective bioagents against *P. tritici-repentis*. The *B. velezensis* strains BZR 336 g and BZR 517 exhibited antagonistic activity against *P. tritici-repentis* Kr-15/2016 in vitro: they inhibited the mycelium growth by 72.4–94.3% and caused its degenerative changes. The results of the experiment clearly show that the treatment of wheat plants with strains *B. velezensis* resulted in a reduction in the development of infection in the climatic chamber and in the field. The use of BZR 336 g and BZR 517 in a three-year field trial provided 5.0–7.6% additional yield. Our studies confirm that BZR 336 g and BZR 517 have both fungicidal and growth-promoting properties. Consequently, they can be used for preventive treatments of agricultural plants. BZR 336 g and BZR 517 are promising for organic farming where chemical fungicides are prohibited. However, it is necessary to more carefully observe the necessary agrotechnical measures when using living means of biocontrol.

## 6. Patents

Asaturova A.M., Tomashevich N.S., Zhevnova N.A., Homyak A.I., Dubyaga V.M., Pavlova M.D., Kozitsyn A.E., Sidorova T.M. "Biofungicide for protecting crops from diseases and increasing yields"/Patent for invention RUS 2621356 3 December 2015.

**Author Contributions:** Conceptualization, A.A., N.Z., N.T., O.K. and G.V.; methodology, N.Z. and O.K.; validation, A.A. and O.K.; formal analysis, N.T. and N.Z.; investigation, M.P. and N.S.; resources, M.P. and N.S.; data curation, N.T. and N.Z.; writing—original draft preparation, N.Z.; writing—review and editing, A.A.; visualization, A.A.; project administration, A.A. and G.V.; funding acquisition, A.A. and G.V. All authors have read and agreed to the published version of the manuscript.

**Funding:** The research was carried out in accordance with the State Assignment of the Ministry of Science and Higher Education of the Russian Federation within the framework of research on the topic No FGRN-2022-0006 «Development of technologies for integrated protection of agricultural crops, taking into account the immunological characteristics of the variety».

**Institutional Review Board Statement:** Not applicable.

**Informed Consent Statement:** Not applicable.

**Acknowledgments:** We express our gratitude to Aleksan Albertovich Khalafyan for statistical data processing.

**Conflicts of Interest:** The authors declare that they have no conflict of interest.

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
