# Peer review of "Efficacy of New Local Bacterial Agents against Pyrenophora tritici-repentis in Kuban Region, Russia"

_agronomy, doi:10.3390/agronomy12020373_

Round 1
Reviewer 1 Report
The manuscript addresses an important research point and uses a bacterial strain of
Bacillus velezensis to control the tan spot in winter wheat. The result of the study seems to be promising, however there are some comments need to be addressed.
The comments are:
Line 47: "Plant Growth-Promoting" is not italic
Line 49-51: please combine this short paragraph to the previous one
Line 73-77: please provide the correct accession numbers, because it searched them in the GenBank and I did not found any match with these numbers.
Line 94-95: please clarify this statement " The original optimized growth medium was obtained on the Czapek medium for bacteria with corn extract as a nitrogen source, and molasses - as a carbon source ", because Czapek medium is a fungal growth medium not for bacteria.
Line 95-96: why do use different temperature and incubation time for the two strain, I think their cultural characteristics are similar!!!
Line 104: " The strains were incubated for 20 days at 28.0°C. ", why 20 days, is 7 days not enough?
Line 131: " Then we carefully removed epicuticular wax coating from the leaves surface ", please explain how did you remove epicuticular wax from wheat leave, using what tool?
Line 193: " The application rate of liquid cultures of BZR 336g and BZR 517 was 3 and 2 l/ t, respectively ", why the titer of the two isolates is different, also I think the 2 or 3 L for tone is too low??
Line 210: please provide the type of analysis if I t was one way ANOVA or what type did you use?
Line 298: " Arrows show the areas with the release of pigmented ", I don't see any arrows? please add them!
Discussion section: please combine the short paragraphs to give a complete meaning.
In addition to above-mentioned comments, I have a question, the study was implemented during 2013-2016, about 5 years ago, why the authors did not publish the results directly after having them or during a short period, I think the delay could make the results relatively old!
Reviewer 2 Report
The authors of the study titled "Efficacy of new local bacterial agents against
Pyrenophora tritici-repentis in the Krasnodar Region of Russia" found some
antagonistic microorganism strains like B. velezensis BZR 336g 11 and BZR 517 have antagonistic activity against P. tritici-repentis, which is innovative work in the field of biological control. However, there few comments:
1. The manuscript needs English revision carefully.
2. Authors used velezensis BZR 336g 11 and BZR 517 strains for treatment
individually. Why the authors did not combine the two strains for disease
treatment? Combining them may give better results than individual use.
3. In conclusion, P.10, the author said, "Our research has confirmed that 415
velezensis BZR 336g and BZR 517 strains can be an alternative to chemical
fungicides in 416 ecological and organic farming systems, as well as in
integrated pest management." I am asking how the mentioned strains can be
alternatived to chemical fungicides, and their efficiency is less than 40%. I
recommend changing the sentence.
Reviewer 3 Report
Dear author,
I have read your manuscript entitled "Efficacy of new local bacterial agents against Pyrenophora tritici-repentis in the Krasnodar Region of Russia." Your contribution needs very hard work to gain scientific merit and deserve publication. Because of this, I recommend " Reconsider after major revision." Note that additional analyses are necessary to provide support to some conclusions.
I am attaching a Word file to help you understand the main failures of your contribution regarding the scientific content (A) and language use (B). I hope you can find them useful.
